# Community-led change: Progress toward policy, systems, and environmental impacts through the Catalyzing Communities initiative

**Travis R. Moore** ⬤*, **Yuilyn A. Chang Chusan***, **Emily Sanderson, Larissa Calancie, Erin Hennessy, Julia M. Appel, Mary Ulseth, Christina D. Economos**

ChildObesity180, Friedman School of Nutrition Science and Policy, Tufts University, Boston, Massachusetts, United States of America

\* travis.moore@tufts.edu

## Abstract

### Purpose and objectives

This study evaluated progress toward policy, systems, and environmental (PSE) changes through community-led actions supported by the Catalyzing Communities initiative, guided by Stakeholder-Driven Community Diffusion (SDCD) theory. Objectives included assessing community action implementation, evaluating SDCD-informed strategies' role in enhancing capacity, and identifying contextual factors influencing progress towards PSE changes.

### Intervention approach

Catalyzing Communities engaged 23 key partners across eight U.S. communities. These partners led committees of 110 local champions and integrated local insights and resources into community-led actions. Capacity-building strategies supported action prioritization and implementation.

### Evaluation methods

A mixed-methods approach included web surveys and follow-up interviews. Surveys assessed action implementation, while interviews explored impacts and contextual factors. Data were analyzed using thematic analysis, supported by an iteratively developed codebook and double coding.

### Results

Fourteen changemakers completed surveys and ten participated in interviews. They led 82 local champions across six communities in prioritizing and implementing 21 community-informed actions. Actions differed across communities based on local priorities, including enhancing early childhood and school programs, improving food

**Data availability statement:** Data can be found via the OSF repository by following this link: https://osf.io/fhk94/files/osfstorage.

**Funding:** CDE received funding for this research from the JPB Foundation under Grant Agreement GR-2020-2503. The funder had no role in study design, data collection and analysis, decision to publish, or preparation of the manuscript.

**Competing interests:** The authors have declared that no competing interests exist.

access, promoting active living, and fostering community connections. Most actions were ongoing, in progress at the time of data collection, and partially successful in reaching target populations. Capacity-building strategies – such as peer networking, tools like causal loop diagrams, and technical assistance – were critical for building relationships, enhancing systems thinking, and securing funding. Contextual factors, including committee synergy, readiness, organizational capacity, and the impact of COVID-19, shaped progress toward PSE changes.

## Introduction

Community-based interventions continue to be a cornerstone in public health for addressing the complex, multifaceted determinants of health that contribute to disparities across populations [1]. These interventions, which often focus on creating Policy, Systems, and Environmental (PSE) changes, are increasingly recognized as essential for achieving sustainable health improvements at the community level and advancing health equity [2]. The concept of PSE change is grounded in the recognition that individual behavior is significantly influenced by the broader context in which people live, work, and play [3]. Further, interventions that modify the environment to make healthy choices the default option tend to be more sustainable and reach a broader audience than those focused solely on individual behavior change [4]. This approach aligns with socioecological models of health behavior, which propose that multiple levels of influence, including interpersonal, organizational, community, and policy levels, should be considered in the design of public health interventions [5].

The importance of building community capacity as a strategy for promoting PSE changes is well-documented in the literature [6,7]. Community capacity refers to the ability of community members to effectively identify problems, mobilize resources, and implement strategies to improve health outcomes [8]. Research has shown that capacity-building interventions can lead to significant improvements in community health by fostering local ownership, enhancing social capital, and facilitating the sustainability of health initiatives [9]. Moreover, the involvement of community partners in the design and implementation of interventions has been shown to increase the relevance, acceptability, and effectiveness of these efforts [10].

Despite the potential benefits of community-based interventions, their success is often contingent on a range of contextual factors that can either facilitate or impede progress. These factors include the availability of resources, the organizational capacity of local institutions, and the broader economic or political conditions within the community. Additionally, the COVID-19 pandemic has introduced unprecedented challenges for many communities, exacerbating existing inequalities and disrupting traditional modes of social interaction and service delivery. However, the pandemic has also provided opportunities for innovation and has highlighted the importance of addressing systemic inequities through PSE changes [11].

The Catalyzing Communities initiative is designed to enhance capacity for community-led PSE change. Catalyzing Communities convenes community partners,

including key community leaders (hereafter, changemakers) and committees of local champions from various sectors that influence child health in communities across the US. Changemakers refer to designated community leaders who co-led the initiative and provided data for this study, committee members (or local champions) refer to individuals who collaborated in developing and implementing community-led actions (CLAs), and community partners is used as a general term to encompass both groups. Through group model building (GMB), a participatory and community-driven approach to engage communities in eliciting and sharing information about complex systems related to child health [12], committee members, often supported by a backbone organization and led by changemakers, seek to understand and address systemic drivers and impacts of child health, prompting to the prioritization of CLAs towards PSE changes. Changemakers play a dual role as both leaders of committee efforts and strategic partners in the research process. They provide ongoing feedback on committee dynamics, successes, challenges, and needs, ensuring that community perspectives shape the development and implementation of CLAs.

## Purpose and objectives

The purpose of this study is to evaluate the progress towards PSE changes through CLAs and capacity-building strategies that were supported by the Catalyzing Communities initiative, informed by the Stakeholder-Driven Community Diffusion (SDCD) framework. This study aims to address the following research questions: From the perspective of community changemakers, (1) to what extent have communities implemented their CLAs?; (2) how have SDCD-informed strategies contributed to progress toward PSE changes?; (3) what contextual factors have influenced progress toward PSE changes?

To explore these questions, we utilized an explanatory sequential mixed methods approach [13]. This approach involved collecting and analyzing quantitative survey data first, followed by qualitative interviews to provide deeper insights and context, helping to validate and elaborate on the survey findings.

## Intervention approach

Catalyzing Communities is rooted in the SDCD framework, which seeks to promote sustainable PSE changes by uplifting local knowledge and leveraging resources to address systemic issues, ultimately aiming to improve community well-being. The SDCD-informed process (Fig 1) includes convening a multi-sector committee to understand the drivers of a chronic disease trend overtime through group model building sessions (GMB), developing action plans with funding support, implementing and evaluating CLAs, and ensuring sustainability through impact measurement and continued funding efforts [14].

The SDCD theoretical framework includes five main strategies that build on the capacity of community partners, including changemakers and committee members, for community-led PSE change: training (including systems-thinking training such as GMB), peer networking opportunities to foster collaboration, technical assistance (TA), tools (e.g., causal-loop diagrams, communication materials), and seed funding for tacking action and incentives. Progress toward PSE changes from capacity-building strategies follows two potential pathways (Fig 2). In the first, SDCD strategies directly trigger community actions, leading to targeted, community-led interventions that address specific local needs and advance PSE changes. In the second pathway, SDCD strategies enhance key capacities—such as knowledge, engagement, systems thinking, social connections, and information diffusion—often conceptualized as coalition capacities [15]. This enhanced capacity strengthens the committee's ability to support and sustain PSE changes in their communities. This paper analyzes how changemakers view progress toward PSE changes stemming from the Catalyzing Communities initiative. We evaluate the extent to which communities have implemented their prioritized CLAs, the influence of Catalyzing Communities strategies on PSE changes, and the contextual factors within organizations, committees, and communities that altered progress. The findings contribute to the evidence base on community-based interventions, highlighting the need for adaptive, context-sensitive, systems-driven approaches in public health, particularly in light of evolving challenges like the COVID-19 pandemic.

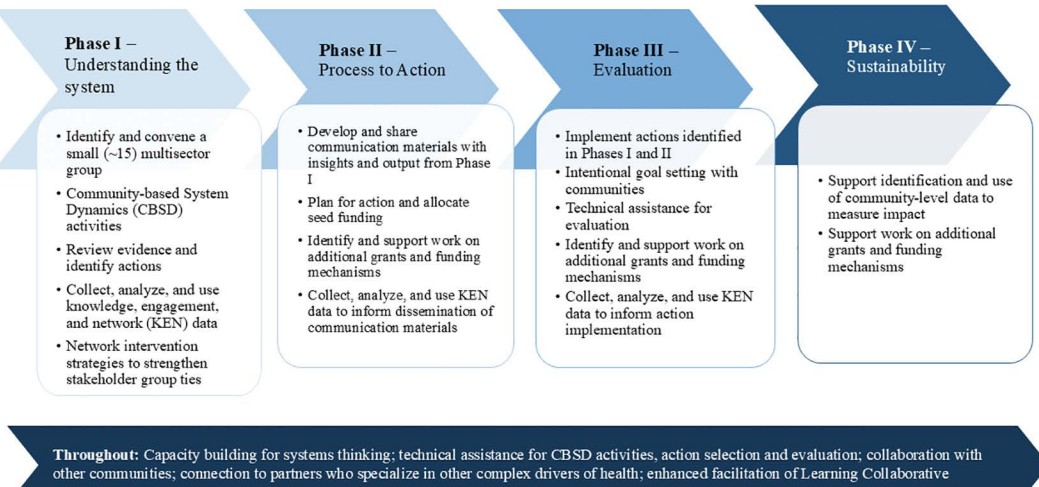

**Fig 1. The Stakeholder-Driven Community Diffusion Phases.** The Stakeholder-Driven Community Diffusion (SDCD)-informed process includes convening a multi-sector committee to understand the drivers of a chronic disease (e.g., obesity) through group model building sessions, developing action plans with funding support, implementing and evaluating community-led actions, and ensuring sustainability through impact measurement and continued funding efforts.

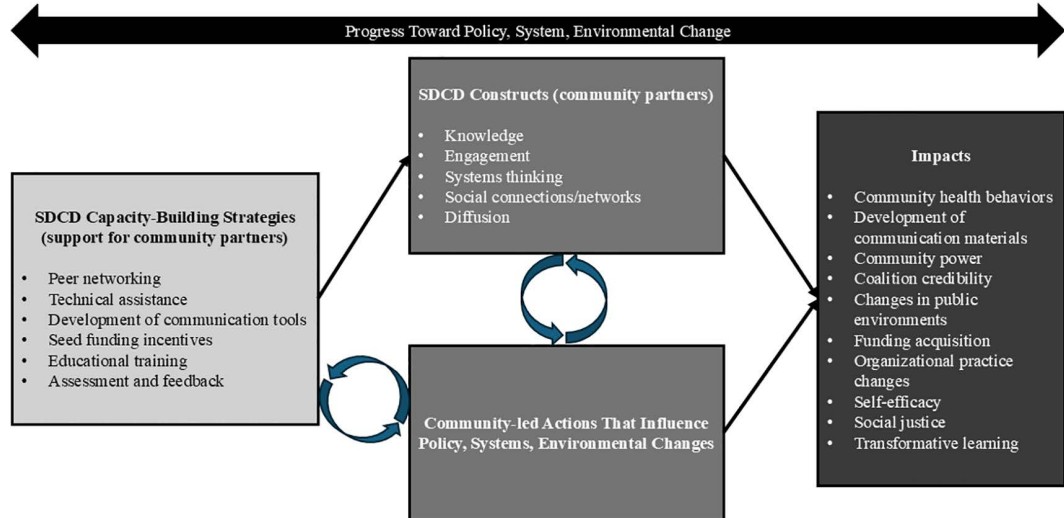

**Fig 2. Study Conceptual Model.** Progress toward policy, systems, and environmental (PSE) changes from capacity-building strategies follows two potential pathways: 1) Strategies informed by the Stakeholder-Driven Community Diffusion (SDCD) theory directly trigger community actions, leading to targeted, community-led interventions that address specific local needs and advance PSE changes; 2) SDCD-informed strategies enhance key community capacities – such as knowledge, engagement, systems thinking, social connections, and information diffusion.

## Methods

### Sample

Catalyzing Communities partnered with key changemakers in eight U.S. communities across two cycles. The first cycle (2018-present) included Cuyahoga County, OH; East Boston, MA; Greenville, SC; Tucson, AZ; and Milwaukee, WI (first

cohort/C1 communities). The second cycle (2023-present) involved Worcester, MA; East Aldine, TX; and Garfield Park, Chicago, IL (second cohort/C2 communities). Community changemakers co-lead the project, offering strategic guidance and ensuring that local needs, insights, and resources are integrated into CLAs. Changemakers were recruited via email for a web-based 15-minute survey and 60-minute follow-up interviews from a sampling frame of 23 changemakers across the eight communities. All participants provided written, informed consent after receiving a detailed explanation of the study's purpose, procedures, and confidentiality protections. Participation was voluntary, and participants were informed that they could decline or withdraw from the study at any time without penalty. The study was approved by the Social, Behavioral, and Educational Research IRB. Data were accessed for research purposes on October 2, 2023, and interviews were conducted during the recruitment period from 10/02/2023–2/01/2024. The authors had access to information that could identify individual participants during and after data collection.

## Surveys and interviews

Surveys, pre-populated with community-specific information, asked changemakers from each study site to list actions they know of that were taken in their communities due to their participation in the project and provided an overview of current initiatives (S1 File: Survey Questions). The survey concluded with a description of potential interview questions. Each individual or group interview included one to three changemakers. Interviews, conducted by a team of five experienced researchers spread across six interviews (S2 File: Interviewers), lasted about 60 minutes each and were audio recorded with participants' consent. The interview protocol included questions on relationships formed, contextual factors influencing progress, changes in work approach, equity promotion, indicators of progress, funding opportunities, and dissemination products (S3 File: Interview Protocol).

## Data analysis

Intensity scores for each PSE change action were calculated from survey responses, capturing four dimensions: impact, completion, duration, and reach, each rated as low, medium, or high, with scores ranging from 1 to 3. Impact levels were categorized as minimal, moderate, or significant; completion from not started to completed; duration from one-time events to ongoing; and reach based on target population engagement (from 0–5% to >20%). Adapted from the Healthy Communities Study [16], this scoring method provided a summary of each community action's characteristics and effectiveness.

Following an explanatory sequential mixed methods approach, quantitative findings were further explored through interviews to clarify and update survey responses as needed. Thematic analysis guided qualitative analysis, with a codebook (S4 File: Codebook) developed iteratively through inductive and deductive approaches, incorporating frameworks like the Evidence-Based System of Innovation Support (EBSIS) by Leeman et al. (2015), Implementation Systems Framework by Wandersman and colleagues (2008), SDCD model, and the socioecological model [17,18]. Coding involved two trained student coders using Dedoose software, ensuring intercoder reliability (Cohen's Kappa $\kappa = .71$) and resolving discrepancies through discussion. Code frequency and co-occurrence tables helped analyze popular themes and connections, and new themes were integrated into the evolving codebook through iterative refinement.

## Results

Fourteen changemakers completed the online survey, and ten participated in online interviews across six communities: Greenville, SC; Tucson, AZ; and Milwaukee, WI; Worcester, MA; East Aldine, TX; and Garfield Park, Chicago, IL, thereby reflecting the perspectives of 82 community leaders across diverse communities in the US. While we do not report changemakers' demographics due to easy identification, Table 1 reports characteristics of the six committees, which were led by 19 changemakers, and the six US communities they represented. Findings are organized into quantitative and qualitative results, drawing from surveys and interviews, respectively.

**Table 1. Community and community partners characteristics.**

| Community | 1 | 2 | 3 | 4 | 5[b] | 6 | 7 | 8 |
|---|---|---|---|---|---|---|---|---|
| **Community characteristics[a]** | | | | | | | | |
| Population estimate (thousands) | 514,213 | 541,482 | 594,548 | 205,319 | 64,670 | 19,992 | 385,282 | 46,655 |
| Land area (mi²) | 785.0 | 226.7 | 96.8 | 37.36 | 20.33 | 1.94 | 82.5 | 4.8 |
| Median household income (USD) | $53,739 | $24,102 | $25,266 | $56,746 | $36,184 | $23,067 | $20,407 | $48,704 |
| Foreign born (%) | 7.9 | 15.3 | 5.0 | 21.9 | 41.8 | 2.1 | 5.9 | 50.4 |
| **Race and ethnicity (%)** | | | | | | | | |
| Hispanic or Latino (all races) | 8.8 | 33.6 | 19.2 | 23.9 | 85.2 | 4.8 | 11.9 | 57.4 |
| NH White | 69.0 | 62.1 | 44.8 | 53.6 | 11.0 | 5.9 | 40.0 | 32.6 |
| NH Black or African American | 18.0 | 5.2 | 38.4 | 12.7 | 2.7 | 84.9 | 48.8 | 2.6 |
| NH American Indian and Alaska Native | 0.2 | 3.7 | 0.8 | 0.5 | 0.9 | 0.1 | 0.5 | 0.0 |
| NH Asian | 2.2 | 3.2 | 4.3 | 6.8 | 0.7 | 0.9 | 2.6 | 3.8 |
| NH Native Hawaiian and Other Pacific Islander | 0.1 | 0.2 | 0.0 | 0.1 | 0.9 | 0.1 | 0.1 | 0.1 |
| NH some other race | 0.1 | 0.1 | 0.2 | 0.1 | 0.8 | 0.1 | 0.1 | 0.2 |
| NH two or more races | 1.7 | 1.6 | 2.4 | 9.8 | 6.9 | 3.3 | 1.8 | 3.4 |
| **Committee characteristics** | | | | | | | | |
| Committee leaders or changemakers[c] (survey/interviews) | 2/1 | 2/2 | 3/2 | 1/1 | 2/1 | 3/3 | 1/0 | 0/0 |
| Committee of local champions, size[d] (n) | 19 | 11 | 13 | 11 | 14 | 14 | 13 | 15 |
| Bachelor's degree and above (%) | 94.7 | 90.9 | 84.6 | 66.7 | 71.4 | 50.0 | 84.6 | 93.3 |
| Female (%) | 78.9 | 90.9 | 76.9 | 81.8 | 71.4 | 100.0 | 84.6 | 66.6 |
| Committee's target populations for PSE change (age) | 0–18 y | 0–18 y | 0–5 y | 0–5 y | 0–18 y | 0–18 y | 0–8 y | 0–18 y |

[a]American Community Survey, 2019. [b]American Community Survey, 2020. [c, d]Across the 8 communities, a total of 110 local champions were led by 3, 2, 3, 3, 5, 3, 2 and 2 changemakers, in each respective community, for a total of 23 changemakers convened in Catalyzing Communities. Changemakers participated in surveys (n = 14) and/or interviews (n = 10). Changemakers who participated in this study led 82 local champions across 6 communities.

Findings on the extent to which communities implemented their prioritized actions and the resulting impacts on PSE changes are based on survey data, which are contextualized with illustrative quotes from the interviews. Interview data further highlights the CLAs derived from the Catalyzing Communities initiative and the SDCD-informed strategies that supported these CLAs, along with the contextual factors that either facilitated or hindered progress toward PSE changes.

### Community actions: types, extent of implementation, and impacts

Table 2 reports quantitative findings, which includes descriptions of: (a) types of CLAs; (b) the extent of CLAs implementation: duration, completion, and reach, considering the time communities had for their implementation; and (c) ratings of CLAs' impacts and intensity of effectiveness towards PSE changes. Across all communities, changemakers listed 21 CLAs and rated the extent of their implementation.

The types of actions were diverse and tailored to the specific priorities of each community. For instance, in one community that focused on food security, partners developed educational materials to support other key community actors (i.e., food security coalition members). In another community committee that was focused on promoting health among children and their families, partners organized community baby showers to build a sense of community and connect families to resources with a goal of addressing high maternal and infant morbidity rates among Black residents.

The extent of community actions captures the breadth and depth of efforts by community partners in implementing various actions to address their local priorities. It assesses how these actions are planned, executed, and reach their intended audiences, offering a comprehensive view of community engagement. The description of the extent of community actions includes *Community actions' duration*, *level of completion*, and *reach.* In making sense of differences of extent

Table 2. Community-led actions (CLAs) influencing policy, systems, and environmental (PSE) changes.

| Community | Community focus areas | Community-led actions (CLAs) | Time for completion of CLAs (months) | Duration (score) | Completion (score) | Reach (score) | Impact (score) | Overall intensity score |
|---|---|---|---|---|---|---|---|---|
| **Cohort 1 communities** | | | | | | | | |
| Community 1 | • Policy, practice, and environmental change<br>• Health equity<br>• WIC participation<br>• Nutrition security and food justice<br>• Health equity | Convene Race Relations Subcommittee | 37 | Ongoing (3) | Completed (3) | Does not apply | Significant (3) | 3.00 |
| | | Evaluate Farmer's Market acceptance of food and nutrition assistance programs | | Multi-day (2) | Completed (3) | Partially successful (2) | Moderate (2) | 2.25 |
| | | Support for initial Food Equity Action Board (FEAB) | | Ongoing (3) | Started/In Progress (2) | Partially successful (2) | Significant (3) | 2.50 |
| | | Integrate language justice in organizational (non-profit) practices | | Ongoing (3) | Started/In Progress (2) | Partially successful (2) | Moderate (2) | 2.25 |
| Community 2 | • Improve school programs that increase access to healthy foods and physical activity opportunities<br>• Increase use of state tax credits for school funding<br>• Youth mental health | Support advertising campaign to increase school contributions under a Public School Tax Credit | 33 | Ongoing (3) | Completed (3) | Partially successful (2) | Significant (3) | 2.75 |
| | | Improve and increase use of Zen Dens to manage stress in partnering Elementary School | | Ongoing (3) | Completed (3) | Partially successful (2) | Moderate (2) | 2.50 |
| | | Support evaluation of stress strategies curriculum | | Ongoing (3) | Started/In Progress (2) | Not yet successful (1) | Moderate (2) | 2.00 |
| Community 3 | • Improve health status of children 0–5 by increasing resource coordination across the community<br>• Advocacy for healthy environments | Provide financial and organizational support for community baby showers | 32 | Multi-day (2) | Started/In Progress (2) | Highly successful (3) | Moderate (2) | 2.25 |
| | | Provide financial and organizational support for a community garden crawl | | Multi-day (2) | Completed (3) | Highly successful (3) | Moderate (2) | 2.50 |
| | | Identify and secure additional funding | | Multi-day (2) | Completed (3) | Highly successful (3) | Significant (3) | 2.75 |
| | | Increase the size and reach of the network | | Ongoing (3) | Started/In Progress (2) | Partially successful (2) | Moderate (2) | 2.25 |
| | | Create an advocacy platform and training | | Ongoing (3) | Started/In Progress (2) | Partially successful (2) | Moderate (2) | 2.25 |
| **Cohort 2 communities** | | | | | | | | |
| Community 4 | • Decrease food insecurity through mitigating impact of cliff effect<br>• Increase equitable funding distribution | Provide financial support to a graphic/web designer to help create a cliff effect toolkit | 9 | One-time (1) | Started/In Progress (2) | Not yet successful (1) | Moderate (2) | 1.50 |
| | | Fund an administrative role at the Worcester Affordable Housing Coalition | | Ongoing (3) | Started/In Progress (2) | Partially successful (2) | Significant (3) | 2.50 |
| | | Hire a local graphic designer/photographer to create a presentation about addressing the root causes and complexity of food insecurity | | Ongoing (3) | Started/In Progress (2) | Not yet successful (1) | Moderate (2) | 2.00 |

*(Continued)*

**Table 2.** (Continued)

| Community | Community focus areas | Community-led actions (CLAs) | Time for completion of CLAs (months) | Duration (score) | Completion (score) | Reach (score) | Impact (score) | Overall intensity score |
|---|---|---|---|---|---|---|---|---|
| Community 5 | • Community-wide, multi-sector movement focused on access to and knowledge of healthy food<br>• Support active lifestyles<br>• Promote community belonging | Provide financial support to the East Aldine Management District as a back bone organization | 8 | One-time (1) | Started/In Progress (2) | Partially successful (2) | Significant (3) | 2.00 |
| | | Support the Revitalizing East Aldine Community Health (REACH) Initiative launch event | | One-time (1) | Completed (3) | Partially successful (2) | Moderate (2) | 2.00 |
| | | Support the development of REACH initiative monthly challenges | | Ongoing (3) | Started/In Progress (2) | Not yet successful (1) | Moderate (2) | 2.00 |
| Community 6 | • Increase access to and consumption of healthy foods through produce prescription programs<br>• Increase community connection through communal meals | Provide financial support for the organization and implementation of Communal Meals | 7 | Ongoing (3) | Started/In Progress (2) | Partially successful (2) | Minimal (1) | 2.00 |
| | | Provide financial support to a grant writer to assist in applying for grants | | Ongoing (3) | Started/In Progress (2) | Not yet successful (1) | Minimal (1) | 1.75 |
| | | Provide financial support to committee participants to promote participation | | One-time (1) | Started/In Progress (2) | Not yet successful (1) | Minimal (1) | 1.25 |

Scoring rubric to calculate intensity of policy, systems, and/or environmental (PSE) influence score based on community actions' (a) Duration, 1 = One-time; 2 = Multi-day; 3 = Ongoing. (b) Reach, 1 = Not yet successful; 2 = Partially successful; 3 = Highly successful. (c) Completion, 1 = Not started; 2 = Started/In progress; 3 = Completed. (d) Impact, 1 = Minimal; 2 = Moderate; 3 = Significant. Scores for each dimension are averaged for each community action.

Table 2 legend: Across all six communities, a total of 21 community-led actions (CLAs) were implemented, most of which focused on food access, early childhood programming, and community engagement. The majority of CLAs were ongoing or multi-day efforts, reflecting sustained commitment to local priorities. Overall, CLAs demonstrated moderate to high levels of completion and impact, particularly in Cohort 1 communities that had longer implementation timelines.

of implementation of actions across communities, we examined the time that each community had to implement their actions. C1 communities had approximately three years between action prioritization and survey completion, whereas C2 communities had less than one year. This difference influenced the extent and depth of the actions undertaken, affecting their overall implementation levels.

***Community actions: Duration.*** This category refers to whether an action was a one-time event, multi-day/multi-year effort, or ongoing initiative. Understanding duration was key to assessing the sustainability and impact of these actions. One-time events can lay the groundwork for future efforts, while ongoing initiatives enable continuous community engagement and iterative improvements.

More than half of the community actions (13 out of 21) were ongoing, such as recurring events or integrating support into existing practices. Multi-day or multi-year actions (4 out of 21) were more common in C1 communities, where additional time since action prioritization facilitated repeated implementation and adjustments. One-time actions (4 out of 21) included financial support to committee members (in addition to participation stipends) or specific outreach events, such as presentations to disseminate information about the committee's work and were more common in C2 communities due to lesser time for repeated actions.

***Community actions: Level of completion.*** This category measures the progress of community actions, from not started or planning, to in progress, or completed. It reflects how far communities have advanced in implementing their prioritized actions, contributing to PSE changes.

Of the 21 community actions, 14 were in progress at the time of data collection and 7 had been completed. All but one of the completed actions occurred in C1 communities.

***Community actions: Reach.*** Reach measures how successfully community actions engaged their intended target populations, classified as not yet successful, partially or highly successful. The target populations included community residents and individuals in institutions or coalitions.

Most community actions (11 out of 21) were partially successful, respondents self-reported that they reached 5%–20% of the intended population. Six actions had not yet reached their intended population (<5%), often due to the limited time since implementation or dependence on completing initial phases. Only one C1 community had three actions that were highly successful, reaching over 20% of the target population.

Table 3 highlights representative quotes from changemakers that illustrate the extent of CLA implementation, organized by duration, completion, and reach. These quotations contextualize the quantitative findings reported in Table 2.

In addition to the extent of implementation of community actions, changemakers reported the perceived impact of their CLAs on PSE changes within the framework of the SDCD theory-informed intervention strategies. The progress towards PSE changes is categorized into *Community actions' impacts*, and *intensity of community actions.*

***Community actions: Impacts.*** This category describes the extent to which CLAs influenced PSE changes. Of the 21 actions, 12 were perceived by changemakers to have a moderate influence, mainly by enhancing skills and services that support PSE progress, such as connecting sectors or sharing resources to improve child health. These actions often aimed to strengthen organizational practices or alter the physical environment in ways that support healthier communities. Significant influence was attributed to 6 actions, which increased access or reduced barriers to PSE changes. Examples include forming a subcommittee on racial equity, establishing a health equity board, and securing grants for future initiatives. These actions were seen as pivotal in advancing community goals and addressing systemic challenges. One community, with the shortest partnership duration (7 months), reported minimal PSE influence for three actions, as they were still in the planning phase. Table 4 presents illustrative quotes describing changemakers' perceptions of the impact of CLAs on policy, systems, and environmental outcomes.

***Intensity of community actions.*** Intensity scores (Table 2) were calculated based on the extent of implementation of community actions and resulting impacts. The average intensity score or effectiveness towards PSE changes for all community actions was medium (2.20) though differences were found across communities from the two cohorts. Among C1 communities, the intensity score ranged from medium to high (2.25–3.00), and in C2 communities the intensity score ranged from low to medium (1.25–2.50). The CLA with the highest intensity score was the formation of a committee focused on racial equity, characterized as a highly impactful and ongoing activity that was completed by the time of survey completion. The CLA with the lowest intensity score was a one-time provision of financial support to foster participation among committee members; although this CLA was already in progress, it was partially successful in reaching committee members by the time of survey completion and it was perceived as having minimal impact.

## Supporting PSE progress through SDCD strategies

CLAs that stemmed from the Catalyzing Communities initiative contributed to PSE change by generating tangible impacts and fostering key SDCD-specific constructs. Across all communities, applying for or securing additional funding was the most frequently mentioned impact (32 mentions), followed by the development of communication materials (31 mentions). Less commonly cited impacts included an increased focus on social justice (24 mentions) and community empowerment (22 mentions), while policy-specific progress was rarely discussed (5 mentions). The most frequently highlighted

**Table 3. Extent of community actions' implementation (duration, completion, and reach), supported by quotes from community changemakers' interviews.**

| Findings | Categories | Illustrative quotes from interviews |
|---|---|---|
| **Extent of Community Actions: Duration, Completion, and Reach.** The extent to which community actions were implemented based on actions' duration, level of completion, and reach was described by community changemakers. Most actions were ongoing/recurring events, that were in progress by the time of data collection, and were partially successful at reaching intended populations. | **Duration of community actions:** refers to the length of the action based on whether community partners perceived it as a one-time, recurrent or ongoing action. | *I think that [the support for a graphic/web designer at the local Community Action Council Center to help create a toolkit about the Cliff Effect, we are] still trying to figure out. It's a little bit we're building the plane while flying it because the Center is just building. At the same time, we're trying to build these resources. I think it's really important [that] we're walking alongside them. (Community 4, C2)* <br> *I would say [supporting advertising campaign to increase schools tax donations is] ongoing because we're working with them all year long, with the tax preparers (Community 2, C1)* <br> *To increase the use of the Zen Den. I think we're continually trying to work with [the Elementary School] to show them ways that [staff and students] can use it and better use it. We're going to continue working with them forever, I think. (Community 2, C1)* <br> *When it comes to the baby shower, each year I think it's getting better. It's improving. Next year will be the fifth year. (Community 3, C1)* |
| | **Completion level of community actions:** refers to community changemakers' perception of how fully an action has been completed based on the steps taken. | *I feel like we have a concrete product we can point to there, where we have a presentation [that shows the complexity of food insecurity] where we never had before a pretty specific thing that we're able to visualize. We're still working on another [presentation]. We have the causal diagrams in this presentation, but we've been working on, is doing this one where we're incorporating narratives and pictures and enhancing it and bringing it to life. That's why I didn't want to put it as like that was complete, totally complete. (Community 4, C2)* |
| | **Reach of community actions:** refers to community changemakers' perception of successfully engaging the intended target population or individuals. | *We went from 0 to 5,400 donations in one school. We made a change in one school out of hundreds. Our goal is really to have a change at the state level on the whole system. (Community 2, C1)* <br> *We don't have a grant writer, I don't think we have a grant writer to assist with applying for more grants. (Community 6, C2)* <br> *We supported [the school staff with] stress strategies implementation, but not the evaluation part. [...] Because that's one of those things that's it's going to take years. (Community 2, C1)* <br> *This year, we had pregnant women come [to the baby shower community event], it didn't start till 10. They came at 9:30, waiting in the lobby. They had so many people come. I know, I had my booth from ten o'clock till one o'clock. I didn't stop, nonstop. (Community 3, C1)* |

C1 = Cohort 1 communities. C2 = Cohort 2 communities.

Table 3 legend: Taken together, these data show that community actions were typically long-term, collaborative efforts that were in progress at the time of data collection and had reached portions of their intended populations. Ongoing actions tended to achieve deeper engagement and higher perceived effectiveness than one-time or short-term initiatives.

SDCD constructs were the development of new social connections (67 times), increased engagement (60 mentions), and enhancement of systems thinking (35 mentions).

While all Catalyzing Community strategies may have played a role in the progress towards PSE changes, and some strategies might be more relevant than others depending on the community, three of these strategies (peer networking, tools, and TA) emerged as the most frequently linked with SDCD Constructs or Impacts. Peer networking emerged as a key contributor to impacts across all communities (67 mentions), followed by the tools used in Catalyzing Communities such as the development and use of causal loop diagrams (40 mentions), and TA from the research team (28 mentions).

Table 5 summarizes co-occurrences between the three most frequently referenced SDCD capacity-building strategies (peer networking, tools, and TA) and the SDCD constructs or impacts they were perceived to be associated with. This table quantifies qualitative linkages identified through thematic analysis. For example, Peer networking was mentioned 67

**Table 4. Illustrative quotes describing perceived CLA impacts on PSE change.**

| Findings | Illustrative quotes from Interviews |
|---|---|
| **Community actions impacts.** In each community, change-makers described their perceived level of impact of community actions. Impact was significant or moderate in C1 communities, and minimal in C2 communities. | *[The advertising campaign to increase tax donations was] significant because [school decision makers] changed the structure of their tax credit. The way that people donate, they can now donate to a tab that is health and wellness tab going forward. That went from zero to something. I think that was for me, significant. (Community 2, C1)* <br> *[The financial support for the committee's backbone organization was] very important because I feel like it set up the committee for long-term success or sustainability because that is now something that's going to be integrated into this organization that is well respected. (Community 5, C2)* <br> *As far as the moderate [rating] for the [support for the launch of the community initiative], I think it's significant to have the launch event happen, but in the long-term impact, it might not be as significant as having the staff person in place. That's why I was rating it [as moderate]. (Community 5, C2)* <br> *[Hiring a grant writer has had] minimal impact because we were still in the planning phase. (Community 6, C2)* <br> *"We hadn't [implemented the action] yet. I tend to think of impact as after things have occurred, how did it play out, what do we see has changed. (Community 5, C2)* |

Table 4 legend: The perceived impacts of community actions primarily involved strengthening relationships, improving organizational practices, and enhancing access to resources. Cohort 1 communities reported more advanced PSE outcomes, while Cohort 2 communities showed early-stage progress reflecting their shorter engagement period.

**Table 5. Code co-occurrence frequencies between capacity-building strategies and SDCD constructs or impacts.**

| Capacity-building strategies (code frequency) | SDCD constructs[a] or impacts[b] (co-occurrence frequency with capacity-building strategies) |
|---|---|
| **Peer networking** (67) | New social connections[a] (44) |
| | Engagement[a] (15) |
| | Knowledge[a] (9) |
| | Community empowerment[b] (9) |
| | Diffusion[a] (6) |
| **Tools** (40) | Systems thinking[a] (12) |
| | Communication materials[b] (11) |
| | Knowledge[a] (10) |
| | Engagement[a] (10) |
| | Diffusion[a] (7) |
| **Technical Assistance** (28) | Funding[b] (9) |
| | Communication materials[b] (8) |
| | Engagement[a] (8) |
| | Systems thinking[a] (6) |
| | Diffusion[a] (5) |

Co-occurrence of capacity-building strategies offered in Catalyzing Communities with [a]community capacity constructs of the Stakeholder-Driven Community Diffusion (SDCD) theory or [b]Impacts, indicating the potential links between intervention strategies and both proximal and distal intervention outcomes. Three strategies were selected as these were the most frequently mentioned strategies by community changemakers.

times across all communities, and it co-occurred with New social connections 44 times, indicating that peer networking strategies may have an influence on establishing new social connections.

Considering the most frequent co-occurrences, this theme is organized into three categories that showcase the role of key SDCD Capacity-Building Strategies in promoting community impact: *Peer networking and social connections*, *tools and systems thinking*, and *technical assistance and funding*.

*Peer networking and social connections.* Peer networking primarily fostered "social connections" (44 co-occurrences), a key SDCD construct, by creating new or strengthening existing relationships that provided social support, information exchange, and a sense of community. Activities like convening regular committee meetings and Learning Collaboratives were instrumental in building these connections. Learning Collaboratives, which were led by the research team with input from a sub-group of community changemakers, brought community partners together monthly and played a crucial role in sustaining the SDCD intervention by facilitating shared learning and collaboration on community priorities.

*Tools and systems thinking.* Tools (e.g., practical communication resources, systems maps, frameworks) were closely associated with the construct of "systems thinking" (12 co-occurrences). Digital collaboration platforms like MURAL and visual tools like causal loop diagrams helped partners visualize complex systems and understand interconnections within their communities. For example, building causal loop diagrams in MURAL allowed partners to see how different factors interacted, supporting decision-making.

*Technical assistance and funding.* TA was linked with applying or securing additional funding (9 co-occurrences). TA, led by the research team with support from community-based system dynamics consultants, included activities, like facilitating meetings, reviewing evidence, and providing feedback on action implementation. TA was particularly instrumental in guiding partners through the process of seeking grants, which directly impacted their ability to continue and expand their initiatives.

Table 6 presents quotes that illustrate the link between SDCD capacity-building strategies and community impacts, providing insights into how these strategies contribute to progress toward PSE changes.

Fig 2 provides a concise visual summary of how SDCD capacity-building strategies connect to community-level impacts, complementing the detailed qualitative findings presented in Tables 5 and 6.

## Contextual factors alter PSE progress

External influences on community progress toward implementing community actions or PSE changes included interpersonal dynamics (13 mentions), organizational factors within changemakers' own institutions or their partner organizations (4 mentions), and the COVID-19 pandemic (4 mentions). Based on these findings, contextual factors were grouped into three categories: *Committee synergy and readiness*, *institutional infrastructure*, and *impact of COVID-19*.

*Committee synergy and readiness.* Pre-existing connections and diverse perspectives among community partners facilitated action development but sometimes hindered alignment on priorities. In one community, it was described how some committee members were used to doing downstream service work, like addressing food security through more food pantries and food box distribution, whereas after working together it was clearer the importance of addressing upstream factors like housing, transportation, childcare, loan forgiveness. Maintaining engagement of the full committee in the intervention was also difficult in some communities, potentially hindering progress.

*Organizational policies and infrastructure.* Limited funding and staffing affected communities' ability to implement and sustain actions. Catalyzing Communities provided TA for grant applications, but backbone organizations' fiscal reliance and staffing shortages hindered progress, even with TA support.

*Impact of COVID-19.* The pandemic disrupted committee activities and limited relationship-building opportunities. However, COVID-19 also highlighted systemic issues and some communities intensified their focus on health disparities, accelerating efforts toward PSE changes.

Table 7 presents illustrative quotes that offer insight into the contextual factors at the interpersonal, institutional, and macro levels, which influenced the trajectory of community actions and, subsequently, progress toward PSE changes.

Table 6. Contributions of SDCD strategies to PSE changes, supported by quotes from community changemakers' interviews.

| Findings | Categories | *Illustrative quotes from interviews* |
|---|---|---|
| **Supporting community capacity and impact through SDCD strategies:** Across all communities, partners described constructs and impacts, considered markers towards PSE changes, within the context of the SDCD capacity building strategies. The most common links between strategies and impacts were: Peer networking and social connections, tools and systems thinking, and technical assistance and funding. | **Peer networking and social connections:** refers to the co-occurrence of the SDCD strategy of bringing committee members together to learn from each other and committee partners' newly established or strengthened social and/or professional relationships. | *Having a chance to connect with community members or community leaders, and really talk about where there are barriers, where there are opportunities to move forward. It started the relationships that allowed us to build many of these initiatives that are going strong now. (Community 1, C1)* <br> *It's been really great to sort of get to know some of the work that [one committee member's organization] does and get to know them. I would say for [my organization], we already had connections with some of the other organizations that are represented in the Committee. This was a way to deepen those relationships. (Community 6, C2)* <br> *Everywhere I go, I'm letting them know what's going on. I've got [business] cards. The idea that I'm part of an eight cities Learning Collaborative, people are like, "Wow, I want to hear more." "I want to be involved. I want to stay connected." I think that made a difference. (Community 3, C1)* |
| | **Tools and systems thinking:** refers to the co-occurrence of the SDCD strategy of pre-planned educational and/or skill-building sessions and committee partners' understanding of complex phenomena, inter-actions, and interrelationships among various components within the system. | *It was a big deal to do that [systems] map. Some people felt like, "Wow, this is overwhelming." At the end when they saw it, they thought, "Wow, this is for real. This is really what happens". It made a big difference on how they looked at the whole picture. It wasn't just one piece, that there's a whole system that needs to be dealt with. I think dealing with it as a collective versus being by yourself and trying to deal with it by yourself is totally different. When we communicate that map to other people, they see the possibilities. They see that now there's system points and places that we can interact. (Community 3, C1)* <br> *I've always been a big picture person and I always want to do all the stuff, but I didn't ever take the time that we took to really look and see how are things connected and how can you intervene in the system, and how can you share that with people to show them that yes, what you're doing and what they're doing are interconnected, and if you work together on coming up with a solution in here, you can fix these issues. For me, that really helped in the work that I do and all the community organizations that I sit on and looking at things in a much different way. (Community 2, C1)* <br> *We always went back to see what our progress was, showing the mural boards that we did, and out of that came this. We can actually see progress from the causal loop diagrams to now there's a web web page where people can learn about [our community actions] and learn how to participate. (Community 5, C2)* |
| | **Technical assistance and funding:** refers to the co-occurrence of the SDCD strategy of interactive support that is individualized to the specific needs of individuals or teams and committee partners ability to identify, apply for, or secure additional funding for ongoing community actions. | *Most of [the] grants that I listed, we would not have been eligible for in the way that we did before our work with Catalyzing Communities. I think having concrete action plans, really having a strong understanding of where we wanted to go as a result of our strategic planning process, put us on a path that we could seek out funding opportunities that were a fit within the work that we wanted to do. I think it put us in a place where federal opportunities open back up, and two, it put us in a place where we were competitive for several [funding] opportunities. (Community 1, C1)* <br> *We applied for [a funding opportunity]. I don't think we went as far in that one but [there] was another proposal we submitted with the Catalyzing Communities team. (Community 4, C2)* |

Table 6 legend: Peer networking, tools such as causal loop diagrams, and TA were consistently linked to improved systems thinking, stronger social connections, and increased access to funding—key pathways through which SDCD strategies contributed to PSE progress.

## Discussion

### Advancing understanding of community-led PSE changes

Community-based interventions targeting PSE changes are essential for addressing social determinants of health and advancing health equity [19]. Although generally effective, research has lacked insights into the mechanisms of community capacity building and their role in sustaining PSE changes. Catalyzing Communities, and the SDCD-informed intervention used in it, addresses this gap by enhancing community capacity and establishing a structured approach to foster

**Table 7. Influence of contextual factors on progress toward PSE changes, supported by quotes from community changemakers interviews.**

| Findings | Categories | Illustrative quotes from interviews |
|---|---|---|
| **Contextual factors**: Factors that preceded or were outside of the Catalyzing Communities initiative spanned across the socioecological levels, and impacted progress towards community actions or PSE changes. | **Committee synergy and readiness:** refers to interpersonal factors within the committee that were either a barrier or facilitated committee synergy and readiness for community actions or PSE changes. | *[We had] opportunities to get engaged with WIC in an earlier opportunity and things like that. As a result of that, two of [the community actions] really started from those discussions. (Community 1, C1)*<br>*One contextual factor is that people are very used to and the habit of have got a pretty well-worn pathways of doing downstream service work, so like addressing food security through more food pantries and food box distribution and whatever, hot meals programs, or holiday giveaways, things like that. I think that we can point to root causes that are upstream factors or living conditions. It's like, yes, housing, transportation, childcare, loan forgiveness – those are food security issues. (Community 4, C2)*<br>*I think we may have been able to have more progress if the full committee was committed. (Community 6, C2)* |
| | **Organizational infrastructure:** refers to funding and personnel infrastructure of backbone or partnering organizations. | *The biggest challenge [to seek funding] I think for us is that we're not a nonprofit and we have a fiscal agent. It is hard to describe that and have [grantors] understand (Community 2, C1)*<br>*One of [our community actions is] going to take years of working with those teachers [in our partnering organization] and then they've had so much change in the teachers. (Community 2, C1)* |
| | **Impact of COVID-19:** refers to the challenges and facilitators inflicted by the pandemic | *It was a little bit more difficult to build relationships. I always believe when you're really working on things that you need to have in person. It's so easy on Zoom to just turn off the camera or not have your camera on (Community 2, C1)*<br>*I think the timing of our planning process and our work with y'all hitting right as COVID hit and there were shutdowns and such strong needs for community support in the area of food security, it really helped this work just explode (Community 1, C1)* |

lasting PSE changes. This study demonstrates how community-based, SDCD-informed strategies such as peer networking, technical assistance, and systems-thinking tools can build community capacity and advance PSE changes. We identified clear patterns linking community engagement strategies with tangible outcomes in relationship-building, resource mobilization, and sustainability, by combining mixed-methods data across six communities.

While the Catalyzing Communities initiative demonstrated promising progress toward PSE changes, several challenges emerged that helped contextualize these findings. Many CLAs were still in early or mid-implementation stages, and several had only partially reached their intended populations by the time of data collection. These patterns reflect both the ambitious nature of PSE work and the short implementation window available for some communities, particularly those in Cohort 2. Additionally, the COVID-19 pandemic disrupted coalition meetings, strained organizational capacity, and limited opportunities for in-person relationship-building—factors that changemakers identified as slowing progress. Recognizing these challenges provides a more balanced view of community trajectories and underscores the importance of sustained support, flexible timelines, and adaptive strategies to translate early systems and environmental gains into longer-term policy change.

### Extending the existing literature

The Catalyzing Communities initiative extends previous research by empirically demonstrating key capacities (e.g., social connections and systems thinking), strategies, and CLAs that may support effective PSE change. Drawing from Leeman et al.'s (2015) adapted EBSIS framework, and other frameworks, this study demonstrates how the SDCD framework can facilitate the practical application of these capacities and strategies, as well as support CLAs, across diverse communities with different socio-political and economic contexts, highlighting factors that influence the sustainability of community-based interventions [17].

## Mechanisms of change and adaptation

A key contribution of this study is its detailed exploration of the potential mechanisms through which strategies and CLAs lead to PSE changes. While previous research has identified the importance of coalition-building and stakeholder engagement [20], this study goes further by mapping out the specific pathways through which these mechanisms may operate. For example, the study highlights how, from the perspective of community changemakers, peer networking strategies strengthen engagement with other community partners and with important child health equity topics and research, ultimately leading to making progress toward PSE changes. Further, mentions of peer networking co-occurred with impacts in policy, community empowerment, and self-efficacy. This focus on the 'how' of community capacity-building and impacts provides insights into the processes that underpin successful community interventions.

Additionally, the study's adaptability in the face of the COVID-19 pandemic underscores its novelty. While the pandemic posed significant barriers to traditional methods of community engagement, the initiative adapted by leveraging digital tools and remote collaboration strategies. This pivot not only ensured the continuation of the project but also highlighted the resilience of the SDCD model and its applicability in crisis contexts. This aspect of the study adds a new dimension to the literature, illustrating how community-based interventions can remain effective even under unprecedented challenges [21].

## Addressing gaps in the literature: community decision-maker perspectives

This study uniquely contributes to the literature by focusing on the perspectives of community changemakers as decision-makers in assessing the progress and impact of PSE changes. While much of the existing literature centers on community member or interventionist perspectives, this study provides valuable insights into how research community partners/changemakers perceive the effectiveness and sustainability of PSE changes. This perspective is crucial, as it highlights the alignment (or misalignment) between community priorities and the strategic objectives of health interventions [22]. Understanding this dynamic is essential for designing more effective community-based interventions that can gain support from both grassroots participants and leadership.

## Limitations

While the Catalyzing Communities initiative sought to advance PSE changes, few direct local- or state-level policy impacts were observed during the study period. This finding likely reflects several intersecting factors operating within complex policy processes. First, policy change often requires extended timelines beyond the scope of our data collection, particularly for communities that had less than one year of implementation (Cohort 2). Second, partners often prioritized actions that could produce visible systems or environmental outcomes (such as new partnerships, improved coordination, or expanded programs) before pursuing formal policy reform. Third, many changemakers operated within organizational or municipal contexts where decision-making authority was distributed or constrained, limiting their ability to directly influence formal policy shifts at local or state levels. Nevertheless, there was some qualitative evidence of organizational-level policy changes in one community (Table 4, Community 2, C1). These findings align with prior research indicating that governance structures and competing local priorities can delay the translation of systems-level insights into formal policy outputs [23]. Future evaluations should continue to track whether early capacity-building and systems changes eventually lead to sustained policy outcomes at various governance levels over time.

Although this study involved diverse communities across the United States, the number of changemakers who completed surveys (n = 14) and interviews (n = 10) was relatively small, reflecting the intentional design of the Catalyzing Communities initiative to engage a select group of community leaders in depth. As such, the findings are not intended to be statistically generalizable but instead offer rich, contextually grounded insights into how community-driven processes unfold within different contexts. The emphasis on depth rather than breadth strengthens the transferability of findings to other multi-sector, systems-oriented community initiatives that share similar structures and goals. Future research with larger samples, including committee members and broader stakeholders, could further validate and extend these findings.

The findings rely primarily on self-reported data from changemakers, which may introduce subjectivity or recall bias. Although these reports provide valuable, insider perspectives on community and implementation processes, future research could strengthen validity by incorporating complementary data sources (e.g., independent policy or program audits, network analyses, or quantitative tracking of PSE outcomes) to triangulate and extend these findings.

### Practical implications and future research directions

The findings from the Catalyzing Communities initiative have several implications for public health practice. The project potentially provides a scalable framework that can be adapted to other communities seeking to implement PSE changes by illustrating structured capacity-building and strategic engagement models that emphasizes systems thinking [12]. Future research should explore the long-term impacts of these interventions and investigate how different community contexts influence the trajectory, adoption, and sustainability of PSE changes. Moreover, there is a need for more research on the role of digital tools in enhancing community capacity and engagement, especially as the pandemic accelerated the shift toward virtual collaboration. To support community actions towards PSE changes, long term community-based research support is critical as a longer time working with communities translates into greater magnitude of PSE changes.

## Supporting information

**S1 File. Survey Questions.**
(DOCX)

**S2 File. Interviewers.**
(DOCX)

**S3 File. Interview Protocol.**
(DOCX)

**S4 File. Codebook.**
(DOCX)

## Acknowledgments

We would like to express our deepest gratitude to the community changemakers who generously shared their insights and experiences. Your dedication and commitment to improving local conditions have been invaluable to and are the foundation of this work.

## Author contributions

**Conceptualization:** Travis R. Moore, Erin Hennessy, Julia M. Appel, Mary Ulseth, Christina D. Economos.

**Data curation:** Yuilyn A. Chang Chusan, Emily Sanderson, Larissa Calancie, Mary Ulseth.

**Formal analysis:** Yuilyn A. Chang Chusan, Emily Sanderson.

**Funding acquisition:** Erin Hennessy, Julia M. Appel, Christina D. Economos.

**Investigation:** Travis R. Moore, Yuilyn A. Chang Chusan, Larissa Calancie, Julia M. Appel, Mary Ulseth.

**Methodology:** Travis R. Moore, Yuilyn A. Chang Chusan, Larissa Calancie, Erin Hennessy, Julia M. Appel, Christina D. Economos.

**Project administration:** Yuilyn A. Chang Chusan.

**Resources:** Christina D. Economos.

**Supervision:** Travis R. Moore.

**Visualization:** Travis R. Moore, Yuilyn A. Chang Chusan.

**Writing – original draft:** Travis R. Moore, Yuilyn A. Chang Chusan.

**Writing – review & editing:** Emily Sanderson, Larissa Calancie, Erin Hennessy, Julia M. Appel, Mary Ulseth, Christina D. Economos.

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
