## [Decision Letter · Decision Letter 0]

7 Oct 2025

Dear Dr. Moore,

Thank you for submitting your manuscript to PLOS ONE. After careful consideration, we feel that it has merit but does not fully meet PLOS ONE’s publication criteria as it currently stands. Therefore, we invite you to submit a revised version of the manuscript that addresses the points raised during the review process.

We look forward to receiving your revised manuscript.

Kind regards,

Alon Harris

Academic Editor

PLOS ONE

Journal Requirements:

When submitting your revision, we need you to address these additional requirements

3. We note that you have indicated that there are restrictions to data sharing for this study. PLOS only allows data to be available upon request if there are legal or ethical restrictions on sharing data publicly. For more information on unacceptable data access restrictions, please see http://journals.plos.org/plosone/s/data-availability#loc-unacceptable-data-access-restrictions .   

Reviewers' comments:

Reviewer's Responses to Questions

**Comments to the Author**

1. Is the manuscript technically sound, and do the data support the conclusions?

Reviewer #1: Yes

Reviewer #2: Yes

2. Has the statistical analysis been performed appropriately and rigorously?

Reviewer #1: Yes

Reviewer #2: I Don't Know

3. Have the authors made all data underlying the findings in their manuscript fully available?

Reviewer #1: Yes

Reviewer #2: Yes

4. Is the manuscript presented in an intelligible fashion and written in standard English?

Reviewer #1: Yes

Reviewer #2: Yes

Reviewer #1: This is a well-structured and important study that explores how the Catalyzing Communities initiative, guided by the Stakeholder-Driven Community Diffusion (SDCD) theory, supported community-led efforts to implement policy, systems, and environmental (PSE) changes. The authors used a mixed-methods approach (surveys and interviews) across eight U.S. communities to evaluate the implementation, capacity building, and contextual influences of these interventions.

The manuscript addresses a timely and critical topic: community-driven, equity-focused change in public health. The explanatory sequential design (quantitative then qualitative) is well-executed and enhances the depth of understanding. Integration of intensity scoring provides a meaningful way to assess the impact of community actions.

The authors did a great job of detailing community-specific differences, especially across Cohort 1 (C1) and Cohort 2 (C2) sites. They account for contextual challenges (e.g., COVID-19, organizational infrastructure) that affect real-world implementation.

Areas for Improvement:

1. Clarity in Writing- Consider streamlining sections of the Results and Discussion for better flow and readability. In particular, tables with large amounts of data (e.g., Tables 2–6) could be supplemented with clearer summaries or synthesis in the text.

2. Despite the paper’s focus on policy, systems, and environmental change, few concrete policy-level impacts were observed or described. A discussion of why policy progress was limited (e.g., timeline, governance barriers) would strengthen the analysis.

3. The communities involved are diverse, but the sample size of changemakers (n=14 surveys, 10 interviews) is relatively small. The authors might consider discussing limits to generalizability or transferability of findings.

4. The manuscript uses “changemakers,” “community leaders,” “committee members,” and “partners” somewhat interchangeably. For clarity, please standardize terminology throughout the manuscript. Consider abbreviating lengthy phrases like “community-led actions” after the first use.

This is a solid and well-designed manuscript. With some editorial tightening, clarification of methodology, and more critical reflection on limitations, it will make a valuable contribution to public health and implementation science literature.

Reviewer #2: Thank you for the opportunity to review this manuscript. The study addresses an important and timely topic, and I commend the authors for their community-engaged approach and clear commitment to equity. I have just a few suggestions that I believe could strengthen the paper:

1. A shorter summary of key findings and simplified tables/figures would help readers quickly understand the main results.

2. Since most findings are based on changemakers’ self-reports, it would be helpful to highlight this more clearly in the limitations and discuss how future work might incorporate more objective measures.

3. Some tables are difficult to follow due to formatting and length. Consider streamlining or supplementing with simple visuals (charts or infographics) to make findings more accessible.

4. The discussion could acknowledge challenges (e.g., partial reach, incomplete actions, COVID-19 disruptions) more explicitly to provide a balanced picture alongside the successes.

Good luck!

**Do you want your identity to be public for this peer review?** For information about this choice, including consent withdrawal, please see our Privacy Policy

Reviewer #1: **Yes: ** Anna Fabczak-Kubicka, MD

Reviewer #2: No

---

## [Author Response · Author response to Decision Letter 1]

10 Oct 2025

Reviewer 1

1. Clarity in Writing- Consider streamlining sections of the Results and Discussion for better flow and readability. In particular, tables with large amounts of data (e.g., Tables 2–6) could be supplemented with clearer summaries or synthesis in the text.

a. We thank the reviewer for this helpful suggestion. We have revised the Results and Discussion sections to improve readability and flow. Specifically, we streamlined descriptions of the quantitative results and added concise synthesis paragraphs summarizing key findings from Tables 2–6. In addition, we included explicit summary statements (or table legends, following style requirements) after each table to help readers quickly grasp the major takeaways from the detailed data presented in the tables.

2. Despite the paper’s focus on policy, systems, and environmental change, few concrete policy-level impacts were observed or described. A discussion of why policy progress was limited (e.g., timeline, governance barriers) would strengthen the analysis.

a. We agree that while our data show meaningful progress toward systems and environmental changes, fewer policy outcomes at the local or state levels were observed. To address this, we have expanded the Discussion section to explicitly discuss potential reasons for limited policy progress, including the relatively short implementation timeline for Cohort 2 communities, the community-led prioritization of actions that focused on visible systems or environmental outcomes, and governance barriers such as limited authority or alignment among community partners; all reflecting the complexity of policy processes at local and state levels We have also highlighted how these contextual challenges shape expectations for policy outcomes in community-based PSE initiatives.

b. Revisions were made in the Discussion to clarify that limited policy progress does not reflect lack of success but rather structural and temporal realities of community-driven change efforts.

3. The communities involved are diverse, but the sample size of changemakers (n=14 surveys, 10 interviews) is relatively small. The authors might consider discussing limits to generalizability or transferability of findings.

a. While the number of participating changemakers was necessarily limited due to the structure of the Catalyzing Communities initiative (where each community designates a small number of key leaders) we agree that this limits the generalizability of findings. To address this, we have expanded the Limitations discussion to explicitly acknowledge the small sample size and note that our findings are best understood as contextually grounded insights into community processes rather than statistically generalizable results. We have also clarified that the study’s strength lies in its depth of engagement and cross-community comparison, which enhances the transferability of findings to similar community-based, systems-oriented initiatives.

b. Revisions were made in the Discussion section.

4. The manuscript uses “changemakers,” “community leaders,” “committee members,” and “partners” somewhat interchangeably. For clarity, please standardize terminology throughout the manuscript. Consider abbreviating lengthy phrases like “community-led actions” after the first use.

a. We appreciate the reviewer’s attention to terminology and clarity. We have reviewed the manuscript carefully to ensure consistent and precise use of terms. We intentionally use “changemakers,” “committee members,” and “community partners” to reflect distinct but complementary roles within the Catalyzing Communities initiative.

i. Changemakers refer to designated community leaders who co-led the Catalyzing Communities initiative alongside researchers..

ii. Committee members represent the broader local champions engaged in planning and implementing community-led actions.

iii. The term ‘Community partners’ is used as an umbrella term when referring collectively to changemakers and committee members.

b. To improve clarity, we added brief definitional statements when these groups are first introduced in the Introduction and ensured consistent usage throughout. We also abbreviated community-led actions as CLAs after the first use, as suggested.

Reviewer 2

1. A shorter summary of key findings and simplified tables/figures would help readers quickly understand the main results.

a. We appreciate this helpful suggestion. In response to a similar comment from another reviewer, we revised the Results section to include concise summary statements that introduce each table, as well as table legends,, highlighting key findings from Tables 2–6. We also streamlined the tables by refining headers and reducing redundant information to improve readability. These changes make the main results easier to interpret at a glance while preserving the richness of the mixed-methods findings.

2. Since most findings are based on changemakers’ self-reports, it would be helpful to highlight this more clearly in the limitations and discuss how future work might incorporate more objective measures.

a. We have added text in the Discussion section to acknowledge that the findings are primarily based on changemakers’ self-reported data and therefore may reflect subjective perspectives. We also note that future studies could complement these insights with more objective or externally validated measures of community progress, such as document reviews, policy audits, or independent assessments of PSE change.

3. Some tables are difficult to follow due to formatting and length. Consider streamlining or supplementing with simple visuals (charts or infographics) to make findings more accessible.

a. We appreciate this suggestion and agree that clarity in presentation is important. To improve readability, we adjusted the formatting of longer tables (particularly Tables 3–6) by refining headers, adding spacing for visual separation between entries, and ensuring consistent labeling of cohorts and communities. We also added brief introductory sentences before several tables to orient readers to their purpose and key takeaways.

b. Given that these tables capture rich qualitative data that substantively illustrate the diversity and nuance of changemakers’ experiences, we retained their level of detail to preserve analytic depth. To further enhance accessibility, we have also included brief summary statements and references to figures summarizing overarching findings across communities.

4. The discussion could acknowledge challenges (e.g., partial reach, incomplete actions, COVID-19 disruptions) more explicitly to provide a balanced picture alongside the successes.

a. We agree with the reviewer that a balanced discussion should include both accomplishments and challenges experienced by communities during implementation. We have expanded the Discussion to explicitly acknowledge barriers such as limited reach among target populations, incomplete or ongoing actions at the time of data collection, and disruptions caused by the COVID-19 pandemic.

---

## [Decision Letter · Decision Letter 1]

28 Oct 2025

Community-led change: Progress toward policy, systems, and environmental impacts through the Catalyzing Communities initiative

PONE-D-25-18403R1

Dear Dr. Moore,

We’re pleased to inform you that your manuscript has been judged scientifically suitable for publication and will be formally accepted for publication once it meets all outstanding technical requirements.

Kind regards,

Alon Harris

Academic Editor

PLOS ONE

Additional Editor Comments (optional):

Reviewers' comments:

Reviewer's Responses to Questions

**Comments to the Author**

Reviewer #1: All comments have been addressed

Reviewer #2: All comments have been addressed

2. Is the manuscript technically sound, and do the data support the conclusions?

Reviewer #1: Yes

Reviewer #2: Yes

3. Has the statistical analysis been performed appropriately and rigorously?

Reviewer #1: Yes

Reviewer #2: I Don't Know

4. Have the authors made all data underlying the findings in their manuscript fully available?

Reviewer #1: Yes

Reviewer #2: Yes

5. Is the manuscript presented in an intelligible fashion and written in standard English?

Reviewer #1: Yes

Reviewer #2: Yes

Reviewer #1: The authors have been highly responsive to reviewer feedback, resulting in a significantly strengthened and more coherent manuscript. The integration of quantitative and qualitative findings is now clearer. The revisions improve both methodological transparency and interpretive depth.

The study offers a meaningful contribution to the literature on community-led, systems-oriented public health initiatives and illustrates the practical application of the SDCD framework. The work’s emphasis on context, capacity building, and community agency provides valuable insights for both researchers and practitioners.

Reviewer #2: (No Response)

**Do you want your identity to be public for this peer review?** For information about this choice, including consent withdrawal, please see our Privacy Policy

Reviewer #1: **Yes: ** Anna Fabczak-Kubicka

Reviewer #2: No

---

## [Editor Report · Acceptance letter]

PONE-D-25-18403R1

PLOS ONE

Dear Dr. Moore,

I'm pleased to inform you that your manuscript has been deemed suitable for publication in PLOS ONE. Congratulations! Your manuscript is now being handed over to our production team.

Kind regards,

on behalf of

Dr. Alon Harris

Academic Editor

PLOS ONE